# Impact of COVID-19 on Subclinical Placental Thrombosis and Maternal Thrombotic Factors

**DOI:** 10.3390/jcm11144067

**Published:** 2022-07-14

**Authors:** Marie Carbonnel, Camille Daclin, Morgan Tourne, Emmanuel Roux, Mathilde Le-Marchand, Catherine Racowsky, Titouan Kennel, Eric Farfour, Marc Vasse, Jean-Marc Ayoubi

**Affiliations:** 1Department of Obstetrics and Gynecology, Hospital Foch, 40, Rue Worth, 92150 Suresnes, France; camille.daclin@wanadoo.fr (C.D.); cracowsky@bwh.harvard.edu (C.R.); jm-ayoubi@hopital-foch.com (J.-M.A.); 2Medical School, University of Versailles, Saint-Quentin-en-Yvelines, 55, Avenue de Paris, 78000 Versailles, France; m.tourne@hopital-foch.com; 3Department of Pathology, Hospital Foch, Hospital Foch, 40, Rue Worth, 92150 Suresnes, France; 4Department of Clinic Research, Hospital Foch, 40, Rue Worth, 92150 Suresnes, France; e.roux@hopital-foch.com (E.R.); m.le-marchand@hopital-foch.com (M.L.-M.); t.kennel@hopital-foch.com (T.K.); 5Department of Obstetrics, Gynecology and Reproductive Biology, Brigham and Women’s Hospital, Boston, MA 02115, USA; 6Department of Clinical Biology, Hospital Foch, 40, Rue Worth, 92150 Suresnes, France; e.farfour@hopital-foch.com (E.F.); m.vasse@hopital-foch.com (M.V.); 7UMR-S 1176, Institut National de la Santé et de la Recherche Médicale, Université Paris-Saclay, 94270 Le Kremlin-Bicêtre, France

**Keywords:** COVID-19, pregnancy, placental vascular pathology, thrombotic, protein Z, ZPI, estradiol

## Abstract

Background: In the context of the SARS-CoV-2 pandemic, our interest was to evaluate the effect of COVID-19 during pregnancy on placenta and coagulation factors. Methods: a prospective cohort study between January and July 2021 of 55 pregnant women stratified into: Group O, 16 patients with ongoing SARS-CoV-2 infection at delivery; Group R, 21 patients with a history of SARS-CoV-2 infection during pregnancy but who recovered prior to delivery; Group C, 18 control patients with no infection at any time. All women had nasopharyngeal SARS-CoV-2 RT-PCR tests performed within 72 h of delivery. Obstetrical complications were recorded and two physiological inhibitors of coagulation, protein Z (PZ) and dependent protease inhibitor (ZPI), were analyzed in maternal and cord blood. All placentae were analyzed by a pathologist for vascular malperfusion. Results: No patient in any group had a severe COVID-19 infection. More obstetrical complications were observed in Group O (O: *n* = 6/16 (37%), R: *n* = 2/21 (10%), C: *n* = 1/18 (6%), *p* = 0.03). The incidence of placental vascular malperfusion was similar among the groups (O: *n* = 9/16 (56%), R: *n* = 8/21 (42%), C: *n* = 8/18 (44%), *p* = 0.68). No PZ or ZPI deficiency was associated with COVID-19. However, an increased ZPI/PZ ratio was observed in neonates of Group R (O: 82.6 (min 41.3–max 743.6), R: 120.7 (29.8–203.5), C: 66.8 (28.2–2043.5), *p* = 0.04). Conclusion: COVID-19 was associated with more obstetrical complications, but not an increased incidence of placental lesions or PZ and ZPI abnormalities.

## 1. Introduction

The coronavirus disease 2019 (COVID-19) caused by the severe acute respiratory syndrome coronavirus 2 (SARS-CoV-2) is a major public health challenge that rapidly spread around the world [1]. In the general population, the main complication of COVID-19 is an acute respiratory distress syndrome [2]. The infection also increases the risk of venous thromboembolism [3] caused by an exaggerated inflammation response called the cytokine storm.

The association of SARS-CoV-2 infection and the prothrombotic state of pregnancy due to estrogens may result in a major risk of thrombotic events [4]. This might be added to other thrombotic risk factors, such as history of thromboembolic disease, thrombophilia or obesity [4]. SARS-CoV-2 infection during pregnancy is associated with more maternal and fetal complications, including preeclampsia (OR 1.6), intrauterine growth restriction (OR 9), preterm delivery (OR 1.48) and stillbirth (OR 2.36) [5,6,7].

These obstetrical complications involve coagulation disorders and placental vascular pathologies [8]. There is an increased thrombin generation because of activation of a coagulation cascade in the maternal circulation due to pathological processes, including inflammation and depletion of anticoagulation proteins. Protein S, protein C and protein Z are implicated in the inhibitory control of the coagulation cascade. Protein Z (PZ) is a cofactor of a serpin, the protein Z-dependent protease inhibitor (ZPI), and the PZ-ZPI complex inhibits activated factor X and has an important anticoagulant role [9]. Therefore, a deficiency in PZ or ZPI induces a prothrombotic state in general population. An altered plasma PZ concentration is also associated with adverse pregnancy outcomes (miscarriage, stillbirth, intra uterine growth restriction and preeclampsia) [10,11,12,13]. Furthermore, ZPI has an anti-inflammatory role, and there is a major increase in the level of ZPI in response to inflammation [14]. As a result, an increased ratio of ZPI/Z suggests an inflammatory state.

The above coagulation changes could lead to thrombosis and vascular abnormality of the placenta at the maternal–fetal interface. Indeed, the villous trophoblast expresses heparin sulfate, protein C and protein Z on its surface [8]. Thrombotic events of placental vessels can cause an impairment of placental perfusion, leading to fetal growth restriction, preeclampsia and fetal death. Lesions on the maternal side include placental abruption and lesions related to maternal malperfusion (increased intervillous fibrin deposition, villous infarcts…). Lesions on the fetal side include vascular lesions consistent with fetal thrombo-occlusive disease, such as thrombosis of the chorionic plate, hypo-vascular and avascular villi [15,16]. Analysis of placentae from patients infected with SARS-CoV-2 showed histopathologic findings of thrombosis, infarcts and vascular wall remodeling, suggestive of placental hypoperfusion and inflammation [17,18,19], but did not necessarily lead to adverse pregnancy outcomes [19]. Studies also revealed rare cases of placental infection without necessarily vertical infection to the fetus [20,21].

In this prospective cohort study, we analyzed the impact of SARS-CoV-2 infection during pregnancy and during delivery on placenta histology and coagulation by studying the PZ-ZPI complex. The primary outcome was the occurrence of placental lesions, and the secondary outcome was the decrease in levels of PZ or rate of ZPI.

## 2. Materials and Methods

### 2.1. Study Approval and Registration

We designed a prospective cohort study, the MaterCov Study, in the Obstetric Department of Foch hospital (Suresnes, France). The study was conducted in accordance with the principles of the Declaration of Helsinki, as well as French statutory and regulatory laws, and received approval from the Institutional Review Board of Ouest V Rennes in December 2020 (IRB number 2020-A03115-34), and clinical trial N° NCT04726111. Patients received information about research to be performed on their biological samples and provided written informed consent to participate. Partners also provided consent for neonate clinical data and cord blood assays.

### 2.2. Participants and Data Collection

The MaterCov study included pregnant women between January and July 2021. Inclusion criteria were pregnant women carrying a singleton and with healthcare insurance, who were admitted for delivery after 20 weeks of gestation (WG). They were systematically tested with a SARS-CoV-2 RT-PCR by nasopharyngeal swab, and IGG anti SARS-CoV-2 serology was performed within 72 h before delivery. The type of SARS-CoV-2 variant was not researched. Exclusion criteria were infection by human immunodeficiency virus, toxoplasmosis, rubeola, cytomegalovirus or syphilis during pregnancy, as well as infection by influenza virus within 10 days prior to delivery. Patients under 18 years or patients under guardianship were also excluded. For the control group, patients with comorbidities, such as diabetes, preeclampsia and high blood pressure, were also excluded. We obtained 3 groups: Group O for patients with ongoing infection during delivery; Group R for patients who developed SARS-CoV-2 infection during pregnancy and who recovered before delivery; and Group C for control patients, who had never had COVID-19.

Maternal data concerning SARS-CoV-2 infection during pregnancy were collected from a survey administered prior to inclusion or were extracted from medical records. Medical records were reviewed using the hospital’s computerized database for collection of the following variables: age, body mass index (BMI) and co-morbidities before pregnancy including BMI > 30 kg/m^2^, asthma, diabetes, high blood pressure (HBP), smoking, medical and obstetrical past history, parity and RT-PCR testing for SARS-CoV-2 virus. Variables were also collected regarding positive testing for SARS-CoV-2 virus during pregnancy, the presence of COVID-19 symptoms, severity (defined as pulmonary embolism, intensive care unit (ICU) hospitalization or maternal death), routine laboratory screens, treatments, hospitalization, maternal and fetal complications (stillbirth, HBP, preeclampsia, threat of premature delivery (TPD), intra uterine growth restriction (IUGR), mode of delivery and maternal death. Neonate variables collected included birthweight, pH at birth, presence of respiratory distress, infection, transfer to the neonatal intensive care unit (NICU) and neonatal death. One month after delivery, patients were contacted by phone and responded to a few questions about their health and that of their newborn.

### 2.3. Biological and Placenta Analysis

Peripheral blood samples were collected within the 24 h before delivery, and cord blood samples and SARS-CoV-2 RT-PCR testing of tissue on both the maternal and fetal sides of the placenta were collected at birth (Figure 1). Biological samples were analyzed by the biology department of Foch Hospital. SARS-CoV-2 RT-PCR testing was performed using Alinity M SARS-CoV-2 RT-PCR assay; Abbott Molecular, Des Plaines, IL, USA. SARS-CoV-2 antibody detection was conducted with the SARS-CoV-2 IgG II Quant assay, (Abbott). Maternal peripheral serum concentrations of estradiol and progesterone were performed (Chemiluminescent Microparticle Immunoassays, Abbott). The expressions of PZ and ZPI were measured by ELISA assays (ZYMUTEST™ Protein Z ELISA kit, HyphenBioMed Neeeuville sur Oise, France and Human Serpin A10/ZPI DuoSet ELISA, R&D Systems, Abingdon, UK) in maternal and cord samples. Usual values of PZ (5th–95th percentiles) in healthy controls were 0.9 to 3.5 µg/mL and 50% to 158% for ZPI, and ZPI/PZ ratio was 28 to 89. We considered that PZ level < 0.9 µg/mL and protein ZPI < 50% constituted a risk factor for a pro-thrombotic state [22], while a ZPI/Z ratio ≥ 90 was a sign of inflammation [23]. All placentas underwent histopathological examination by the Anatomical Pathology Department of Foch Hospital and were examined by the same pathologist. If delivery occurred during the day, the placentae were sent fresh and secondarily fixed in formalin; if delivery occurred at night, the placentae were directly fixed in formalin. After a minimum of 24 h of fixation, macroscopic examination was performed to evaluate the membranes, umbilical cord, placental dimensions, basal plate, chorionic plate and appearance of the section slices. For each patient, two umbilical cord specimens, one membrane specimen and five placental specimens, (all formalin-fixed and paraffin-embedded), were studied microscopically by grading the following lesions according to the 2016 Amsterdam Consensus Conference: maternal vascular malperfusion lesions (villous infarction, microthrombi, decidual arteriopathy, retroplacental hematoma) and fetal vascular malperfusion lesions (subchorionic thrombosis, obliteration of chorionic plate vessels and inflammatory remodeling (acute and chronic chorioamnionitis, villitis)), separating maternal and fetal response. Lesions were considered significant when they represented more than 5% of the placental volume. The maternal COVID-19 status was known by the pathologist for safety reasons.

### 2.4. Statistical Analysis

For clinical and biological data, Microsoft Excel software was used for data recording and analyses were performed using SAS v9.4. The quantitative data are described as median values [Interquartile] and categorial variables as frequencies and percentages. For quantitative variables, the non-parametric Kruskal–Wallis tests were used for comparison between groups. For qualitative variables, chi-square tests were used to compare groups or Fisher tests when expected values were below 5. Correlations between variables were calculated using the non-parametric Spearman rank-order test. *p* values < 0.05 were considered significant.

## 3. Results

### 3.1. Demographic and Clinical Features of Study Groups

Fifty-five patients met our inclusion criteria for one of the three study groups and were enrolled to participate; 16 patients had an ongoing COVID-19 infection (RT-PCR SARS-CoV-2 positive at delivery; Group O); 21 patients were RT-PCR SARS-CoV-2 positive during pregnancy but recovered and tested RT-PCR SARS-CoV-2 negative at delivery (Group R); and 18 patients had no history of COVID-19 infection either during their pregnancy or at delivery (RT-PCR SARS-CoV-2 negative anti-SARS -Cov-2 IgG serology negative at delivery). None of the patients were vaccinated against COVID-19.

Demographic and clinical features of study patients are summarized in Table 1. All three groups were comparable for age, BMI, incidence of obesity, gravidity, parity and all assessed comorbidities. Regarding thrombotic risk factors, they appeared to be similar between the groups. No history of thromboembolic disease was mentioned, but one patient in control group had an antiphospholipid syndrome and had taken a prophylactic dose of heparin during pregnancy. Patients with COVID-19 and obesity also had prophylactic doses of heparin during infection. No other infections were observed during pregnancy. The median gestational age at which COVID-19 occurred was 38 WG (30–41) for pregnant women with ongoing infection (Group O) and 27 WG (8–37) for recovered patients (Group R) (*p* < 0.001). All recovered patients had positive RT-PCR SARS-CoV-2 during pregnancy and negative RT-PCR SARS-CoV-2 at delivery. There was no difference between the two groups concerning symptoms; 18 (86%) recovered patients and 13 (87%) patients with ongoing infection were symptomatic. The most common symptoms were anosmia (*n* = 17, 46%), ageusia (*n* = 16, 43%), asthenia (*n* = 16, 43%), rhinorrhea (*n* = 10, 27%) and fever (*n* = 9, 24%).

Only two (12%) patients in Group O and one (5%) patient in Group R (who had no associated comorbidities) required oxygen therapy. No patients had deep vein thrombosis, none were admitted to the ICU and there was no case of maternal death. All patients had a complete recovery. Obstetrical complications were significantly increased in Group O (*n* = 6; 37%) compared with Group R (*n* = 2; 10%) and Group C (*n* = 1; 6%) (*p* = 0.03) (Figure 2). One patient with an ongoing infection had preeclampsia with HELLP syndrome (association of hemolysis, elevated transaminases and low platelet). This patient developed symptoms of COVID-19 at 37 weeks of gestation. She had a normal pregnancy and no comorbidities. She was admitted at 38 weeks for a fever of 39.5 °C, myalgia and asthenia. Blood testing indicated normal hemoglobin (13.2 g/dL), lymphopenia (0.9 G/L), low platelet (60 G/L), cytolysis (9-fold the normal value) and an increased protein C reactive (CRP) (101 mg/L). She had an induction of labor for HELLP syndrome, and finally a C-section for anomaly of the fetal heart rate. Her eutrophic newborn was transferred to the NICU for respiratory distress. Mother and child both had favorable outcomes without sequels of SARS-CoV-2 infection. Another patient in Group O, who had had COVID-19 symptoms for 15 days, was admitted at 29 weeks of gestation for oxygen therapy. She had no associated comorbidities, but blood testing found an increased CRP (22 mg/L) and cytolysis (increased three-fold above the normal value). On the second day of hospitalization, her fetus underwent demise, and the same day she delivered a eutrophic stillbirth (1480 gr), with a placental abruption. The mother achieved complete recovery from SARS-CoV-2 infection. The other complications in Group O were IUGR (*n* = 4, 25%). At birth, the weights of three neonates were less than the tenth percentile and one was less than the third percentile. In Group R, complications were high blood pressure for one patient and IUGR for another one, without consequences for delivery or for the newborn. In the healthy controls (Group C), the sole complication was a threat of preterm birth for one patient, who delivered at term.

The median gestational age at term was similar in all groups (Group O: 39 weeks (range, 29–41); Group R: 40 weeks (range 37–41); and Group C: 39.5 WG (range 37–41). No case of preterm delivery was observed except for the stillbirth. One (6%) patient in Group O required induced labor at 38 weeks and oxygen therapy. However, there was no complication for either mother or newborn at delivery and no clinical signs of infection were reported for any of the neonates.

One month after birth, 33 (60%) mothers answered the phone survey (10 in Group O, 12 in Group R, 11 in Group C). No mother had sequels of SARS-CoV-2 infection or complications due to COVID-19, while two (4%) children were hospitalized for 3 to 8 days for pulmonary infection not related to COVID-19. No cases of SARS-CoV-2 infection or additional complications were reported for the children.

### 3.2. Placenta and Protein Z and ZPI Analysis

Results are summarized in Table 2. Regarding histopathological analysis, the incidence of significant vascular malperfusion was similar for both groups of women infected by COVID-19 compared to the control group (Group O: *n* = 9 (56%); Group R: *n* = 8 (42%) versus Group C: *n* = 8 (44%), *p* = 0.68) (Figure 3). Regarding signs of maternal vascular malperfusion, there was no difference among the three groups in the presence of villous infarct (Group O: *n* = 5 (31%); Group R: *n* = 4 (21%); Group C: *n* = 6 (33%), *p* = 0.68) or microthrombi (Group O: *n* = 6 (37%); Group R: *n* = 6 (32%); Group C: *n* = 4 (22%), *p* = 0.62). There were 5% of vessels affected by microthrombi in Group O and C. Placental analysis of the fetus diagnosed with IUGR (estimated as lower than the 3rd percentile) found two old and marginal villous infarcts of 10 and 15 mm, which represented a volume of 4%, and the presence of microthrombi (5% of affected vessels). The placentae of other two fetuses diagnosed with IUGR (estimated inferior to 10th percentile) in Group O and R exhibited no anomalies. There were no cases of decidual arteriopathy. We reported one recent placental abruption of 43 mm, responsible for the stillbirth described previously. Regarding signs of fetal vascular malperfusion, although more subchorionic thrombosis was observed in both COVID-19 groups than in the control group, there was no statistical difference (Group O; *n* = 3 (19%), Group R: *n* = 3 (16%); Group C: *n* = 0, *p* = 0.11). There was no case of chorionic thrombosis. Regarding signs of inflammation, there were more cases of acute chorioamnionitis in the COVID-19 groups than in the control group, but again there was no statistical difference (Group O: *n* = 3 (19%); Group R: *n*= 1 (5%), Group C: *n* = 0, *p* = 0.10). Placental analysis of patients with preeclampsia found signs of acute chorioamnionitis, classified as stage 2 on the maternal side and stage 1 on the fetal side, without signs of vascular malperfusion [24]. The RT-PCR tests on the placenta were positive for SARS-CoV-2 virus. Placental analysis of the patients with high blood pressure (Group R) and with threat of preterm delivery (Group C) were normal.

The analysis of PZ and ZPI levels (Table 3) revealed no difference in rates of PZ or ZPI in either maternal blood or fetal cord blood among the three groups (PZ in mothers *p* = 0.49, in fetal cord *p* = 0.49; ZPI in mothers *p* = 0.51, in fetal cord *p* = 0.3). However, for the patient with preeclampsia and HELLP syndrome, the level of the PZ was decreased (PZ: 0.75). The ZPI/PZ ratio was significantly higher in neonates of Group R than in Group C (Group R ZPI/Z ratio = 120.7 (29.8–203.5), group C ZPI/Z ratio = 66.8 (28.2–2043.5), *p* = 0.04).

The progesterone concentration in maternal serum was higher in Group R (200.9 ng/mL (86.8–245.0) than in Group O (136.7 ng/mL (35.5–186.8) (*p* = 0.009). The estradiol concentration in maternal serum was also higher in Group R (22,210 pg/mL (10,450–31,760) than in either Group O (16,400 pg/mL (6455–24,590) or C (15,710 pg/mL (7086–50,000) (*p* = 0.007). Estradiol and progesterone were both significantly decreased in patients with ongoing infection, as compared to the recovered and control groups (Figure 4).

A Pearson’s correlation test was performed between the ZPI/Z ratio and the concentrations of progesterone and estradiol for each of the three groups. In contrast to Groups O and C, for which no significant correlations were found (data not shown), this analysis revealed a correlation between the ZPI/Z ratio and the concentration of both estradiol (*p* = 0.03) and progesterone (*p* = 0.04) in newborns of Group R (Figure 5).

## 4. Discussion

This prospective cohort study with pregnant women investigated the association between SARS-CoV-2 infection and obstetrical and perinatal outcomes, and on placental modifications by studying placenta histology and the PZ-ZPI complex.

The incidence of obstetrical complications was significantly increased in those patients with COVID-19 at the time of delivery (Group O: 37%) versus those who had recovered from COVID-19 (Group R: X%) and those who had never tested positive (Group C: Y%). Four patients in Group O developed severe obstetrical complications, including preeclampsia with HELLP syndrome (*n* = 1, 6%), stillbirth (*n* = 1, 6%) and fetal growth restriction (*n* = 4, 25%). Although the number of patients we were able to enroll was relatively small, the incidence of preeclampsia and fetal growth restriction we observed were not unlike those of Villar et al. who reported an 8% risk of developing preeclampsia, a 22.5% risk of premature birth and a 20.5% risk of low birth weight [25]. De Sisto and al. reported an incidence of 1.3% for stillbirth [26]. Of note, the risk of these complications increases in the third trimester [27]. There was no difference concerning delivery and neonatal outcomes.

Pregnant women included in this study did not develop COVID-19 requiring ICU, nor were there any maternal deaths. This is in line with our department’s experience in the first year of the pandemic, in which we observed a low ICU hospitalization rate of 1.2% for COVID-19 patients [28]. One of the explanations for this low rate could be a low incidence of associated comorbidities in this patient population. According to Vivanti and al., pregnant women had an increased risk of developing severe COVID-19 if they had associated comorbidities [29]. An ICU-admission rate of 8.4% and a mortality rate of 1.6% is described in literature regarding pregnant women with COVID-19 [25].

This study did not highlight more placental pathologies in pregnant women infected by SARS-CoV-2. Shortly after the onset of the pandemic, studies described placental vascular malperfusion on both the maternal and fetal sides, including retroplacental hematoma, accelerated villous maturation and intramural fibrin deposition in COVID-19 pregnant patients [19]. These pathological changes were not specific to SARS-CoV-2 infection, and one of the hypotheses was an abnormal uterine perfusion due to maternal hypoxia secondary to severe COVID-19 [30]. Another could be inflammatory changes affecting the placenta [7]. These could lead to clinical preeclampsia, preterm birth, fetal demise and fetal growth restriction [8]. Furthermore, placental pathology was associated with more severe neonatal outcomes, with a higher rate of admission to the NICU [31]. Nevertheless, a recent meta-analysis of 30 studies concluded that the incidence of vascular and inflammatory lesions was comparable to that of non-COVID-19 pregnancies, and there was no specific lesion associated with COVID-19 [32]. These results are consistent with the assumption that SARS-CoV-2 does not directly induce placental injury. Another explanation could be the increased levels of circulating extracellular vesicles induced by COVID-19 [33], which are already known to be involved in the hypercoagulable and pro-inflammatory intravascular reactions during preeclampsia [34].

We observed one case of placental infection with detection of SARS-CoV-2 virus by RT-PCR test on placental tissue, although the neonate showed no signs of infection, allowing us to conclude that there were no signs of vertical transmission. However, the placental analysis revealed signs of chorioamnionitis. Male et al. described perivillous fibrin deposition, histiocytic intervillositis and trophoblast necrosis in infected placentae [7]. There are very few cases describing placental infection in women with severe SARS-CoV-2 infection, and they do not necessarily lead to neonate infection, as the rate is estimated to be between 0.9 and 2.8% [7,20,21]. Taken together, the available data regarding neonatal COVID-19 cases are reassuring: the vast majority of infants do not experience significant morbidity and mortality [35].

To our knowledge, the study we report here is the first to analyze PZ and ZPI levels during COVID-19 in pregnant women. There was no difference in the expression of PZ and ZPI among the three groups. Because of the inflammation state caused by COVID-19 and the activation of coagulation, we should expect a decrease in the PZ levels and an increase in ZPI. A decrease in the level of PZ l was described in cases of thrombosis, suggesting an important anticoagulant role [9]. The levels of PZ in this study were consistent with the clinical data finding no increased risk of thrombosis related to COVID-19 in pregnant women [4,36]. The physiopathological reason for these phenomena is unknown. A decreased PZ-ZPI complex is also described in such obstetrical complications as fetal growth restriction, preeclampsia and stillbirth [11,12,13]. We observed a deficit of PZ in the patient with preeclampsia and HELLP syndrome.

The ZPI/Z ratio can be used as a marker of inflammation because of the excess of production of ZPI in inflammation [14]. In our study, the ZPI/Z ratio was significantly increased in newborns of the recovered group compared to the control group. This could be explained by a potential inflammation state in neonates born to mothers who infected by SARS-CoV-2 during pregnancy. Inflammation states with increased cytokine levels have been previously described in neonates [7].

We also investigated the rate of pregnancy hormones, which have a role in inflammation and prothrombotic states [37]. Progesterone and estradiol levels appeared to be higher in pregnant women who contracted SARS-CoV-2 infection during pregnancy compared to pregnant women with SARS-CoV-2 infection at delivery. It has been reported that higher rates of estradiol and progesterone modulate inflammatory response to infection and have a protective role for women [38].

In this study, we highlighted a link between ZPI/Z ratio in neonates and rates of both estradiol and progesterone in pregnant women from the recovered group. The ZPI/Z ratio decreased when the estradiol and progesterone levels rose. As a result, we can hypothesize that both estradiol and progesterone could have a role in decreasing neonate inflammation states induced by COVID-19. In the literature, there are reports attesting to a protective role of hormones in women [37,38].

One of the strengths of this study was that it was conducted at the beginning of the pandemic and included naïve patients without any vaccination or history of previous COVID-19. This was ideal to better understand the physiopathological mechanisms involved. Indeed, to date, most of the population has either been infected by SARS-CoV-2 or has been vaccinated. Furthermore, the population involved in this study, presenting mainly with mild symptoms, was more representative of the majority of cases of COVID 19 in pregnancy, with, fortunately, only a few severe cases, due to a systematic screening of the patients at the beginning of labor [39]. The histology analysis was performed by one pathologist, which reduced inter-individual biases. This study was also the first study to evaluate the impact of COVID-19 on PZ and ZPI during pregnancy There are several limitations to our study; these include in particular the relatively small number of individuals in each study group, and more particularly, the small number of patients with severe forms of COVID-19. We chose to divide our patients into an Ongoing Infection group and a Recovery group. We did not study the effects of the delay between the beginning of the infection and delivery. The pregnant women were included over a short period of time to limit the potential bias associated with the successive appearance of the different variants. Nevertheless, even in this short period, two variants gained successive dominance: VOC-202012/01 (Alpha variant) and B.1.617.2 (Delta variant), with different impacts on pregnancy, as Delta showed worse outcomes [26]. As the variant was not a focus of this study, this could have induced heterogenous groups. It would be interesting to be able to apprehend them in subsequent studies. Furthermore, other proteins of coagulation, already described in obstetrical complications, could have been studied, such as protein S, protein C, antithrombin and factor V Leiden. The fact that non-blind analysis of the placenta was performed by the pathologist is another limitation.

## 5. Conclusions

To conclude, the MaterCov study found an increased risk of developing obstetrical complications in pregnant women with COVID-19. There were no more placental pathologies in pregnant women infected by SARS-CoV-2 compared with other pregnant women. Placental thrombotic and inflammation lesions were found in patients with obstetrical complications. There was one case of placental infection. We did not highlight a decrease in PZ or ZPI levels.

## Figures and Tables

**Figure 1 jcm-11-04067-f001:**
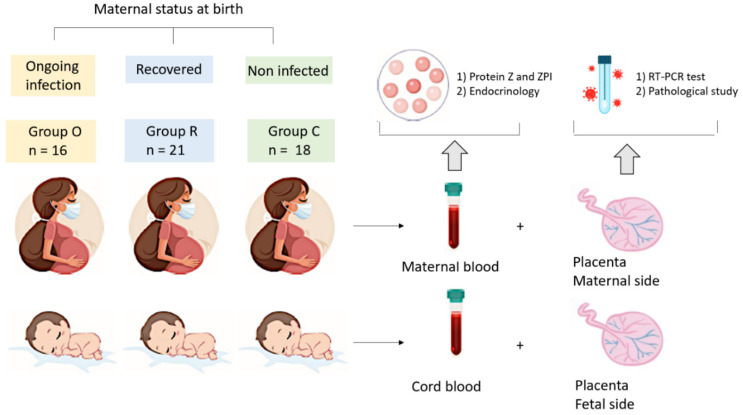
Study outline of the recruitment of neonates and mothers.

**Figure 2 jcm-11-04067-f002:**
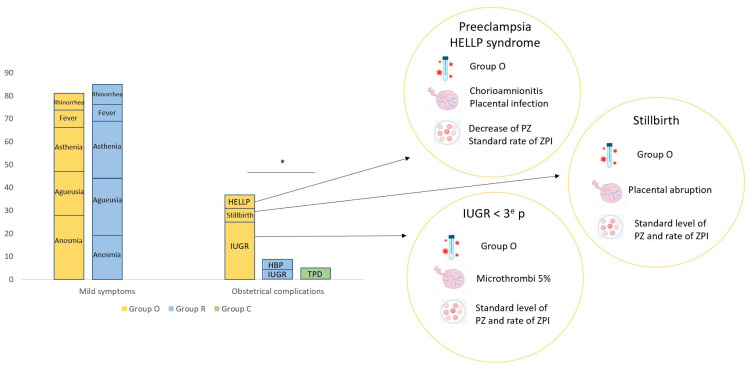
Clinical data of SARS-CoV-2 infection. HELLP: hemolysis, elevated liver enzymes, low platelet count; IUGR: intra uterine growth restriction; HBP: high blood pressure; TPD: threat of premature delivery, * *p* < 0.05.

**Figure 3 jcm-11-04067-f003:**
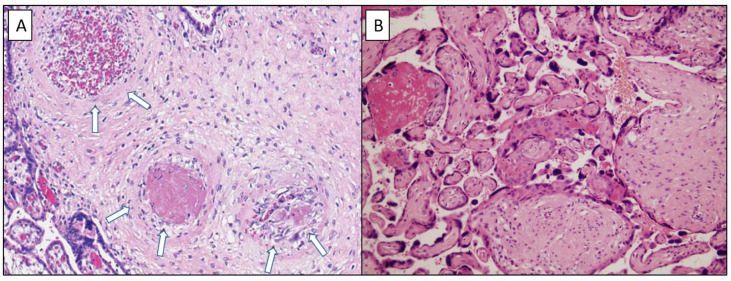
Maternal vascular malperfusion lesions associated with fibrinonecrotic microthrombi at different stages of formation ((**A**): HES ×20) within capillary vascular structures (arrows) and avascular terminal villi ((**B**): HES ×20).

**Figure 4 jcm-11-04067-f004:**
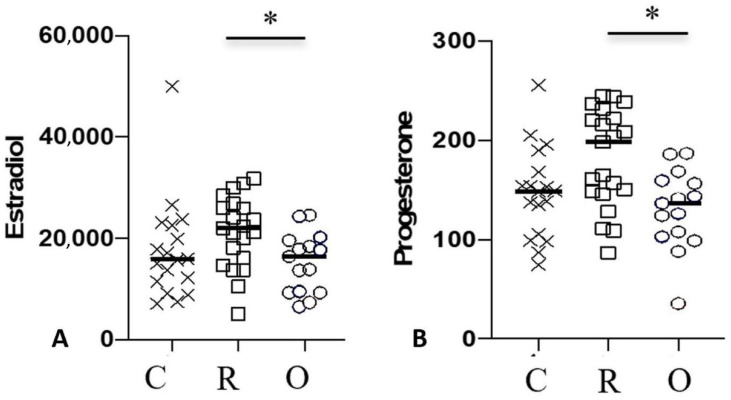
Concentrations of estradiol (pg/mL) and progesterone (ng/mL) in mothers. Data are shown for healthy controls (n = 18; C: crosses), recovered patients (n = 21; R: open squares) and patients with ongoing infection (n = 16; O: circles). Black lines represent the median. * *p* < 0.05.

**Figure 5 jcm-11-04067-f005:**
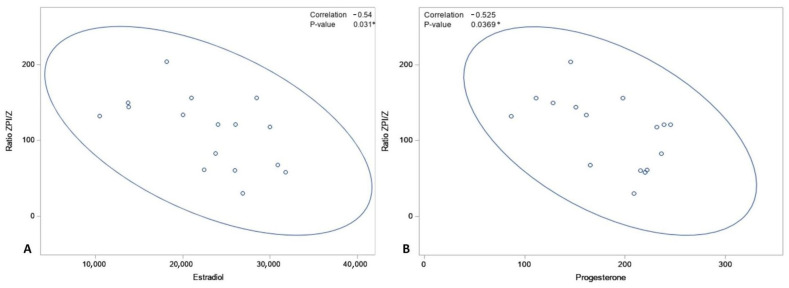
Pearson correlation test between ZPI/Z ratio in neonate and of the concentration of estradiol (**A**) and progesterone (**B**) in mothers, * *p* < 0.05.

**Table 1 jcm-11-04067-t001:** Maternal and Neonatal clinical characteristics.

Maternal SARS-CoV-2 Status	Group O	Group R	Group C	*p* Value
Maternal Characteristics (*n*)	*n* = 16	*n* = 21	*n* = 18	
Median age, years (min–max)	34.0 (21–40)	31.0 (24–43)	32.5 (26–44)	0.52
Median BMI (min–max)	28.4 (21–36)	25.5 (19–31)	27.0 (22–34)	0.09
BMI > 30 kg/m^2^	4 (25%)	2 (9.5%)	4 (22%)	0.29
Gravidity, mean (min–max)	1.0 (1–3)	1.5 (1–7)	1.0 (1–3)	0.65
Parity, mean (min–max)	0 (0–2)	1.0 (0–3)	1.0 (0–3)	0.08
Asthma	0	2 (10%)	2 (11%)	0.54
High blood pressure	0	0	0	NA
History of Preeclampsia	1 (6%)	0	0	0.29
Diabetes	0	0	0	NA
Gestational age of COVID-19 infection Median (min–max)	38 (30–41)	27 (8–37)	NA	**<0.001**
Days between positive SARS-CoV-2+ status (nasopharyngeal swab) and birth, median (min–max)	7 (0–56)	84 (21–225)	NA	**<0.001**
Positive IgG serology at inclusion	8 (50%)	15 (71%)	0	**<0.001**
COVID-19 symptoms	13 (87%)	18 (86%)	NA	1
Fever	4 (25%)	5 (24%)	NA	1
Cough	6 (37%)	9 (43%)	NA	0.74
Asthenia	5 (31%)	11 (52%)	NA	0.20
Anosmia	7 (44%)	10 (48%)	NA	0.81
Ageusia	5 (31%)	11 (52%)	NA	0.20
Rhinorrhea	3 (19%)	7 (33%)	NA	0.46
Asymptomatic	3 (14%)	2 (13%)	NA	1
Severity signs	2 (12%)	1 (5%)	NA	0.56
Hospitalization	3 (19%)	0	3 (17%)	0.09
Hospitalization for COVID-19	1 (6%)	0	0	0.29
Total number of obstetrical complications	6 (37%)	2 (10%)	1 (6%)	**0.03**
Stillbirth	1 (6%)	0	0	0.29
High Blood Pressure	0	1 (5%)	0	1
Preeclampsia	1 (6%)	0	0	0.29
Threat of premature delivery	1 (6%)	0	1 (6%)	0.52
IUGR	4 (25%)	1 (5%)	0	0.10
Term at birth median (min–max)	39 (29–41)	40 (37–41)	39.5 (37–41)	0.68
Pyrexia during labor	1 (6%)	0	0	0.29
**Neonatal characteristics (*n*)**	***n* = 16**	***n* = 21**	***n* = 18**	
Cesarean section (mode of delivery)	3 (19%)	4 (19%)	1 (6%)	0.45
Weight median (g) (min–max)	3100 (1484–4140)	3320 (2760–4270)	3515 (2660–4198)	0.13
pH median (min–max)	7.25 (7.20–7.35)	7.24 (7.08–7.43)	7.23 (7.09–7.45)	0.26
Transfer to ICU	2 (13%)	0	0	0.07
Respiratory distress	1 (6%)	1 (5%)	0	0.75

**Table 2 jcm-11-04067-t002:** Placenta histopathologic analysis.

	Group O *n* = 16	Group R *n* = 19	Group C *n* = 18	*p* Value
Vascular malperfusion	9 (56%)	8 (42%)	8 (44%)	0.68
Maternal side				
Villous infarct	5 (31%)	4 (21%)	6 (33%)	0.68
Volume (%)	3 (2–4)	3 (1–5)	2 (2–5)	0.67
Microthrombi	6 (37%)	6 (32%)	4 (22%)	0.62
Affected vessels (%)	5 (2–5)	5 (2–5)	2 (2–5)	0.19
Deciduous arteriopathy	0	0	0	-
Retroplacental hemorrhage	1 (6%)	0	0	1
Fetal side				
Subchorionic thrombosis	3 (19%)	3 (16%)	0	0.11
Surface (%)	5 (2–20)	2 (1–2)	-	0.16
Chorionic thrombosis	0	0	0	-
Acute chorioamnionitis	3 (19%)	1 (5%)	0	0.10

**Table 3 jcm-11-04067-t003:** Protein Z and ZPI levels.

	Group O	Group R	Group C	*p* Value
**Maternal blood**	***n* = 14**	***n* = 16**	***n* = 18**	
PZ	2.7 (0.6–4.4)	2.3 (0.9–4.4)	2.1 (1.2–4.6)	0.49
ZPI	173.5 (63.0–247.0)	187.5 (89.0–288.0)	164.5 (81.0–276.0)	0.51
Ratio ZPI/Z	64.4 (44.8–216.7)	66.7 (42.7–145.0)	69.9 (38.5–130.7)	0.87
**Fetal cord blood**	***n* = 14**	***n* = 17**	***n* = 17**	
PZ	0.4 (0.1–2.3)	0.4 (0.2–0.7)	0.5 (0.02–0.7)	0.49
ZPI	34.0 (17.0–194.0)	45.0 (17.0–92.0)	35.0 (15.0–55.0)	0.3
Ratio ZPI/Z	82.6 (41.3–743.6)	120.7 (29.8–203.5)	66.8 (28.2–2043.5)	**0.04**

PZ: protein Z; ZPI: protein Z-dependent protease inhibitor.

## Data Availability

The data presented in this study are available on request from the corresponding author. The data are not publicly available due to ethical.

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
