# Peer review of "Impact of COVID-19 on Subclinical Placental Thrombosis and Maternal Thrombotic Factors"

_jcm, 2022, doi:10.3390/jcm11144067_

Round 1

Reviewer 1 Report

The topic of the article is actual, in the context of the Covid-19 pandemic. Although two years of research have been conducted and the results of Covid-19 in terms of vascular disease are far from elucidating. From this point of view, the purpose of the article is very actual and interesting.

The introductory chapter is too long and I think it needs to be redone. The risk factors for hypercoagulability in pregnancies not associated with Covid-19 infection are not mentioned.

In the case of the Method, the authors generally talk about Covid-19 infection without specifying the virus-type that led to the infection. If there were several types, the study groups are heterogeneous. No analyzes have been performed to rule out other pathological conditions that may induce hypercoagulability and thrombosis.

The results are well presented but start from certain wrong premises that can generally make mistakes.

No statistically significant results were obtained regarding PZ or ZPI levels that could lead to the conclusions presented in the article.

Author Response

The topic of the article is actual, in the context of the Covid-19 pandemic. Although two years of research have been conducted and the results of Covid-19 in terms of vascular disease are far from elucidating. From this point of view, the purpose of the article is very actual and interesting.

Point 1: The introductory chapter is too long and I think it needs to be redone. The risk factors for hypercoagulability in pregnancies not associated with Covid-19 infection are not mentioned.

Dear author, thank you very much for your comments

Response 1: We reviewed the introduction to make it drastically shorter and more focused on the main subject of the study. As you suggest we added: Line 50-53 : “The association of SARS-CoV-2 infection and the prothrombotic state of pregnancy due to estrogens may result in a major risk of thrombotic events (4). This might be added to other thrombotic risk factors such as history of thromboembolic disease, thrombophilia or obesity (4). “

Point 2: In the case of the Method, the authors generally talk about Covid-19 infection without specifying the virus-type that led to the infection. If there were several types, the study groups are heterogeneous.

Response 2:

We specified in methods:

Line 101-102” The type of SARS-CoV-2 variant was not research”.

We modified the discussion to show that the groups are heterogeneous due to the two types of dominant variants during the study:

Line 382-388: “The pregnant women were included over a short period of time to limit the potential bias associated with the successive appearance of the different variants. Nevertheless, even in this short period two variants were subsequently dominant : VOC-202012/01 (Alpha variant) and B.1.617.2 (Delta variant) with different impact on pregnancy as Delta has showed worse outcomes (26). As the type of variant was not searched in this study, this could have induced heterogenous groups.”

Point 3: No analyzes have been performed to rule out other pathological conditions that may induce hypercoagulability and thrombosis

Response 3: We didn’t find a significant difference between groups regarding thrombotic risk factors. We added a sentence in results:

L 178-182: “Regarding of thrombotic risk factors, they appeared to be similar between the groups. No history of thromboembolic disease was mentioned, only one patient in control group had an antiphospholipid syndrome and had prophylactic dose of heparin during pregnancy. Patients with COVID-19 and obesity had also prophylactic dose of heparin during infection.”

Point 4: The results are well presented but start from certain wrong premises that can generally make mistakes.

No statistically significant results were obtained regarding PZ or ZPI levels that could lead to the conclusions presented in the article.

Response 4:

We agree that some wrong premises could have biased our results. That’s why in the discussion we emphases the heterogenicity of the groups depending on variants as previously explained in Response to point 2. We questioned the limits due to the design of our study: Line 380-382 “We chose to divide our patients into Ongoing infection group and Recovery group. We didn’t studied effects of the delay between the beginning of the infection and delivery.”

We rewrote the conclusion and removed the part of the sentence concerning the case of preeclampsia as it was not statistically significant:

Line 397-398: “We did not highlight a decrease in PZ or ZPI levels"

Reviewer 2 Report

The manuscript by Marie Carbonnel et al. deals with the impact of SARS-CoV-2 infection during pregnancy and during delivery on placenta histology and coagulation protein Z (PZ) and dependent protease inhibitor (ZPI). The topic is of high interest and the manuscript well written. These are my suggestions to improve the main message.

1)      The introduction is too long and should be focused on the main topic.

2)      Methods: the sentence “For control group, non-infected pregnant women with comorbidities as diabetes, preeclampsia and high blood pressure were excluded” is not clear. Which control group are you referring to? The inclusion criteria and the study groups should be defined

3)      “We considered that PZ level < 0.9 μg/mL and protein ZPI < 50 % constitute a risk 145 factor for a pro-thrombotic state, a ZPI/Z ratio ≥ 90 was a sign of inflammation”, the sentence needs references.

4)      Results: the“incidence of expression” of PZ is not clear, please clarify

5)      Data on thrombrophylaxis with heparin should be provided

6)      Levels of circulating extracellular vesicles might be the link between covid-19 and obstetrical complications. This should be added to the discussion (see for example Campello E. et al Circulating microparticles in umbilical cord blood in normal pregnancy and pregnancy with preeclampsia, 2015 and Longitudinal Trend of Plasma Concentrations of Extracellular Vesicles in Patients Hospitalized for COVID-19, 2022).

Author Response

The manuscript by Marie Carbonnel et al. deals with the impact of SARS-CoV-2 infection during pregnancy and during delivery on placenta histology and coagulation protein Z (PZ) and dependent protease inhibitor (ZPI). The topic is of high interest and the manuscript well written. These are my suggestions to improve the main message.

        Point 1:      The introduction is too long and should be focused on the main topic.

        Response 1: Thank you for your comments. We reviewed the introduction to make it drastically shorter and   more focused on the main subject of the study.

        Point 2:      Methods: the sentence “For control group, non-infected pregnant women with comorbidities as diabetes, preeclampsia and high blood pressure were excluded” is not clear. Which control group are you referring to? The inclusion criteria and the study groups should be defined

Response 2: For the Method, I have rewritten the beginning of “Participants and data collection” to be clearer: control group was the group of patients who never had SARS-CoV-2 infection.

Line 105-109: “For Control group, patients with comorbidities as diabetes, preeclampsia and high blood pressure were also excluded. We obtained 3 groups: Group O for patients with ongoing infection during delivery; Group R for patients who developed SARS-CoV-2 infection during pregnancy and who recovered before delivery; and Group C for control patients, who never had COVID-19”

        Point 3:   “We considered that PZ level < 0.9 μg/mL and protein ZPI < 50 % constitute a risk  factor for a pro-thrombotic state, a ZPI/Z ratio ≥ 90 was a sign of inflammation”, the sentence needs references.

         Response 3: We added two references to support these sentence: Line 153-154: “We considered that PZ level < 0.9 µg/mL and protein ZPI < 50 % constitute a risk factor for a pro-thrombotic state (22), a ZPI/Z ratio ≥ 90 was a sign of inflammation (23).“

  1. Sofi F, Cesari F, Abbate R, Gensini GF, Broze G, Fedi S. A meta-analysis of potential risks of low levels of protein Z for diseases related to vascular thrombosis. Thromb Haemost. avr 2010;103(4):749‑56.
  2. Girard TJ, Lasky NM, Tuley EA, Broze GJ. Protein Z, protein Z-dependent protease inhibitor (serpinA10), and the acute-phase response. J Thromb Haemost. févr 2013;11(2):375‑8.

        Point 4:      Results: the “incidence of expression” of PZ is not clear, please clarify

Response 4: We apologize, the sentence was unclear, we have changed for:  Line   263-264: “The analysis of PZ and ZPI levels (Table 3) revealed no difference in rates of PZ or ZPI in either maternal blood and fetal blood cord among the three groups.”

Point 5:      Data on thromboprophylaxis with heparin should be provided

Response 5: Thank you for this comment which is very relevant, I had data regarding heparin treatment:

Line 178-182: No history of thromboembolic disease was mentioned, one patient in control group had an antiphospholipid syndrome and prophylactic dose of heparin during pregnancy. Patients with COVID-19 and obesity had also prophylactic dose of heparin during infection.

        Point 6:       Levels of circulating extracellular vesicles might be the link between covid-19 and obstetrical complications. This should be added to the discussion (see for example Campello E. et al Circulating microparticles in umbilical cord blood in normal pregnancy and pregnancy with preeclampsia, 2015 and Longitudinal Trend of Plasma Concentrations of Extracellular Vesicles in Patients Hospitalized for COVID-19, 2022).

Response 6:  Thank you for your help, these articles are very interesting, we added a sentence for them : Line 326-329 : “Another explanation could be the increased levels of circulating extracellular vesicles induced by COVID-19 (31), which are already known to be involved in the hypercoagulable and pro-inflammatory intravascular reactions during preeclampsia (32).”

Round 2

Reviewer 1 Report

I have analyzed the changes made to the article and I consider that they meet my expectations.